# Chronic Use of Proton-Pump Inhibitors and Iron Status in Renal Transplant Recipients

**DOI:** 10.3390/jcm8091382

**Published:** 2019-09-03

**Authors:** Rianne M. Douwes, António W. Gomes-Neto, Michele F. Eisenga, Joanna Sophia J. Vinke, Martin H. de Borst, Else van den Berg, Stefan P. Berger, Daan J. Touw, Eelko Hak, Hans Blokzijl, Gerjan Navis, Stephan J.L. Bakker

**Affiliations:** 1Department of Internal Medicine, Division of Nephrology, University Medical Center Groningen, University of Groningen, 9700 RB Groningen, The Netherlands (A.W.G.-N.) (M.F.E.) (J.S.J.V.) (M.H.d.B.) (E.v.d.B.) (S.P.B.) (G.N.) (S.J.L.B.); 2Department of Clinical Pharmacy and Pharmacology, University Medical Center Groningen, University of Groningen, 9700 RB Groningen, The Netherlands; 3Unit PharmacoTherapy, -Epidemiology and –Economics, Groningen Research Institute of Pharmacy, University of Groningen, 9713 AV Groningen, The Netherlands; 4Department of Gastroenterology and Hepatology, University Medical Center Groningen, University of Groningen, 9700 RB Groningen, The Netherlands

**Keywords:** proton-pump inhibitors, iron, iron deficiency, renal transplantation

## Abstract

Proton-pump inhibitor (PPI) use may influence intestinal iron absorption. Low iron status and iron deficiency (ID) are frequent medical problems in renal transplant recipients (RTR). We hypothesized that chronic PPI use is associated with lower iron status and ID in RTR. Serum iron, ferritin, transferrin saturation (TSAT), and hemoglobin were measured in 646 stable outpatient RTR with a functioning allograft for ≥ 1 year from the “TransplantLines Food and Nutrition Biobank and Cohort Study” (NCT02811835). Median time since transplantation was 5.3 (1.8–12.0) years, mean age was 53 ± 13 years, and 56.2% used PPI. In multivariable linear regression analyses, PPI use was inversely associated with serum iron (β = −1.61, *p* = 0.001), natural log transformed serum ferritin (β = −0.31, *p* < 0.001), TSAT (β = −2.85, *p* = 0.001), and hemoglobin levels (β = −0.35, *p* = 0.007), independent of potential confounders. Moreover, PPI use was independently associated with increased risk of ID (Odds Ratio (OR): 1.57; 95% Confidence Interval (CI) 1.07–2.31, *p* = 0.02). Additionally, the odds ratio in RTR taking a high PPI dose as compared to RTR taking no PPIs (OR 2.30; 95% CI 1.46–3.62, *p* < 0.001) was higher than in RTR taking a low PPI dose (OR:1.78; 95% CI 1.21–2.62, *p* = 0.004). We demonstrated that PPI use is associated with lower iron status and ID, suggesting impaired intestinal absorption of iron. Moreover, we found a stronger association with ID in RTR taking high PPI dosages. Use of PPIs should, therefore, be considered as a modifiable cause of ID in RTR.

## 1. Introduction

Iron deficiency (ID) is very common in renal transplant recipients (RTR), with reported prevalence of 20% to 30% more than 12 months after transplantation [1,2,3]. ID is an important contributor to post-transplant anemia, which affects approximately 20% to 49% of RTR within the first year after transplantation and is associated with adverse health outcomes [1,4,5,6]. Besides clinical symptoms associated with ID, such as fatigue, dyspnea, and decreased exercise tolerance, iron deficiency anemia (IDA) has been associated with an increased risk of graft failure and mortality in RTR [4,6,7]. Moreover, iron deficiency, independent of anemia, has been shown to be a risk factor for mortality in RTR [3].

Identifying modifiable risk factors of post-transplant ID may improve transplant outcomes and quality of life in RTR. In this regard, drug-induced factors should not be ignored. Recently, several observational studies have demonstrated that chronic proton-pump inhibitor (PPI) use may negatively affect iron status and is associated with ID in the general population [8,9,10,11]. It is postulated that PPIs interfere with the absorption of iron in the duodenum, where non-heme iron is primarily absorbed in its ferrous form (Fe^2+^) after the reduction from its less absorbable ferric form (Fe^3+^), which is facilitated by gastric acid and membrane reductases localized at the apical membrane of the enterocytes [12,13]. This hypothesis is supported by a study from Ajmera et al., who found a reduced response to oral supplementation of ferrous sulfate in iron deficient patients taking omeprazole [14]. In a large population-based case-control study, an increased risk of ID was found among patients receiving PPI therapy for at least one year and even among intermittent long-term PPI users compared to PPI non-users [8]. These findings are in line with previous results from another large cohort study in the United States, which demonstrated a higher risk of ID among chronic users of both PPIs and H2-receptor antagonists (H2RAs), which diminished after treatment discontinuation [9].

PPIs are frequently prescribed after renal transplantation to prevent gastrointestinal complications from immunosuppressants, and may therefore possibly contribute to the high burden of post-transplant ID in RTR. It is currently unknown whether chronic PPI use adversely affects iron status in RTR and studies investigating this hypothesis are lacking. In the present study, we aimed to investigate the association of PPI use with iron status in a large single-center cohort of stable outpatient RTR.

## 2. Methods

### 2.1. Study Design

For this cross-sectional cohort study, we used data from a previously well-described cohort of 707 stable RTR registered at clinicaltrials.gov as “TransplantLines Food and Nutrition Biobank and Cohort Study”, NCT02811835 [15]. In brief, all adult RTR with a functioning graft for at least 1 year without known or apparent systemic illnesses (i.e., malignancies, opportunistic infections) who visited the outpatient clinic of the University Medical Center Groningen (UMCG) between November 2008 and March 2011 were invited to participate. Written consent was obtained from 707 of the initially 817 invited RTR. Study measurements were performed during a single study visit at the outpatient clinic.

### 2.2. Exposure Definition

RTR using any PPI on a daily basis during a period of at least 3 months before the study visit were defined as chronic PPI users. For statistical analyses we excluded RTR with missing data on PPI dosage (n = 1), with on-demand PPI use (n = 3), with missing data on iron status parameters (n = 7), or using iron supplements or EPO stimulating agents (n = 50), leaving 646 RTR eligible for analysis.

### 2.3. Study Approval

The study protocol was approved by the institutional review board (METC 2008/186, approved on 17 September 2008) of the UMCG and all study procedures were performed in accordance with the Declaration of Helsinki and the Declaration of Istanbul.

### 2.4. Clinical Measurements and Iron Status Parameters

Information on medical history, including reported history of gastritis or peptic ulcer disease, was obtained from electronic patient records as described previously [15]. Medication use, including the use of PPIs, diuretics, renin-angiotensin-aldosterone system (RAAS) inhibitors, antiplatelet drugs, anti-diabetic drugs, mycophenolate mofetil (MMF), calcineurin inhibitors (CNIs) and prednisolone, was recorded at baseline. Blood pressure was measured using a standard protocol, as described previously [16]. Information on alcohol use and smoking behavior was obtained using a questionnaire.

Blood samples were collected after an 8–12 h overnight fasting period. Serum creatinine was measured using an enzymatic, isotope dilution mass spectrometry traceable assay (P-Modular automated analyzer, Roche Diagnostics, Mannheim, Germany). Estimated glomerular filtration rate (eGFR) was calculated applying the serum creatinine-based chronic kidney disease epidemiology collaboration (CKD-EPI) equation. Concentrations of glucose, hemoglobin A1c (HbA1c), and high-sensitivity C-reactive protein (hs-CRP) were determined using standard laboratory methods. Serum iron was measured using photometry (Modular P800 system; Roche Diagnostics, Mannheim, Germany). Serum ferritin concentrations were determined using the electrochemiluminescence immunoassay (Modular analytics E170; Roche Diagnostics, Mannheim, Germany). Transferrin was measured using an immunoturbidimetric assay (Cobas-c analyzer, P-Modular system; Roche Diagnostics, Mannheim, Germany). Transferrin saturation (TSAT, %) was calculated as 100 × serum iron (µmol/L)/25 × transferrin (g/L). Iron deficiency was defined as transferrin saturation (TSAT) < 20% and ferritin < 300 µg/L, as described in literature previously and commonly used in patients with pro-inflammatory conditions, such as chronic heart failure and chronic kidney disease [3,17,18,19]. Proteinuria was defined as urinary protein excretion ≥ 0.5 g/24 h.

### 2.5. Assessment of Dietary Iron Intake

Total dietary iron intake (i.e., heme and non-heme iron) was assessed using a validated semi-quantitative food frequency questionnaire (FFQ), which was filled out at home [20,21]. Dietary data were converted into daily nutrient intake using the Dutch Food Composition Table of 2006 [22].

### 2.6. Statistical Analyses

Statistical analyses were performed using Statistical Package for the Social Sciences (SPSS), version 23.0 (IBM corp., Armonk, NY, USA). Data are presented as mean ± SD for normally distributed data, median with interquartile range (IQR) for skewed data, and number with percentage for nominal data. Differences between PPI users versus PPI non-users were tested using independent sample T-tests, Mann–Whitney U-tests, and Chi-square tests or Fishers exact test when appropriate.

To investigate the association of PPI use with serum iron, serum ferritin, TSAT, and hemoglobin levels, univariable and multivariable linear regression analyses were performed with adjustment for potential confounders of iron status including: age, sex, eGFR, proteinuria, time since transplantation, history of gastrointestinal disorders (i.e., reported history of gastritis or peptic ulcer disease before baseline), lifestyle parameters (BMI, smoking behavior, and alcohol use, dietary iron intake), inflammation (hs-CRP), MMF use, and other medication use (i.e., diuretics, RAAS-inhibitors, anti-platelet therapy, CNI use, and prednisolone use). Serum ferritin was natural log (ln) transformed to obtain a normal distribution. To investigate a dose-response relationship, we performed additional analyses in which RTR were divided into three groups based on daily PPI dose defined in omeprazole equivalents: no PPI, low PPI dose (≤20 mg omeprazole equivalents/day (Eq/d)), and high PPI dose (>20mg omeprazole Eq/d) [23]. Tests of linear trend were conducted by assigning the median of daily PPI dose equivalents in subgroups treated as a continuous variable. To investigate the association between PPI use and ID, we performed logistic regression analyses with adjustment for the same potential confounders used in multivariable linear regression analyses. In sensitivity analyses, H2RA users (n = 20) were excluded to assess the robustness of the association between PPI use and ID. Additionally, we performed sensitivity analyses using an alternative definition of ID as proposed in a position statement by the European Best Practice (ERBP) group and previously recommended in the United Kingdom-based National Institute for Health and Care Excellence (NICE) guideline (NG8) (TSAT < 20% and ferritin < 100 µg/L) [24,25]. A two-sided *p*-value < 0.05 was considered statistically significant in all analyses.

## 3. Results

### 3.1. Baseline Characteristics

Baseline characteristics are shown in Table 1. At baseline, RTR were 53 ± 13 years old and 382 (59.1%) were male. Mean BMI was 26.7 ± 4.8 kg/m^2^, and 157 (24.3%) had diabetes. RTR were included at a median of 5.3 (1.8–12.0) years after transplantation. Mean eGFR was 53.5 ± 19.9 mL/min/1.73 m^2^ and 135 (21.0%) had proteinuria. Mean serum iron and median ferritin concentrations were 15.2 ± 5.9 µmol/L and 115.5 (53.0–213.3) µg/L, respectively. Mean hemoglobin concentration was 13.3 ± 1.7 g/dL and mean TSAT was 25.1 ± 10.5%. Iron deficiency was present in 193 (29.9%) RTR. PPIs were used by a small majority of 363 (56.2%) RTR and omeprazole was the most often prescribed PPI (n = 317). Other PPIs used included esomeprazole (n = 28), pantoprazole (n = 15), and rabeprazole (n = 3). RTR who used PPIs were older than RTR who did not use PPIs, had a higher BMI, and had shorter time between transplantation and baseline measurements. Furthermore, diabetes was more prevalent in RTR using PPIs and PPI users had higher glucose and HbA1c levels, and lower levels of hemoglobin, iron, ferritin, and TSAT. Dietary iron intake was not significantly different between PPI users and PPI non-users. Additionally, CNIs and MMF, diuretics, anti-diabetic drugs, and antiplatelet drugs were more often used by PPI users compared to PPI non-users.

### 3.2. Association of PPI Use with Iron Status Parameters

In univariable linear regression analyses, PPI use was associated with a 2.18 µmol/L lower serum iron (95% CI: −3.09 to −1.27, *p* < 0.001), −0.34 µg/L lower ln serum ferritin (95% CI: −0.49 to −0.18, *p* < 0.001), 3.9% lower TSAT (95% CI: −5.5 to −2.3, *p* < 0.001), and 0.52 g/dL lower hemoglobin levels (95% CI: −0.78 to −0.25, *p* < 0.001). The association between PPI use and lower iron status parameters remained independent of adjustment for potential confounders, as shown in Table 2.

### 3.3. Association of PPI Use with ID

In crude logistic regression analysis, PPI use was associated with ID (OR: 1.95; 95% CI 1.37–2.77, *p* < 0.001), as shown in Table 3. The association remained independent of adjustment for age, sex, eGFR, proteinuria, time since transplantation, and history of GI-disorders (OR: 1.57; 95% CI 1.07–2.31, *p* = 0.02). Further adjustment for lifestyle parameters, including dietary iron intake (OR: 1.57; 95%CI 1.04–2.38, *p* = 0.03) and inflammation (OR: 1.56; 95% CI 1.06–2.30, *p* = 0.03), did not materially affect the association. In model 5 we adjusted for MMF use, which is known for its myelosuppressive nature. In this model, PPI use remained independently associated with ID (OR: 1.57; 95% CI 1.07–2.31, *p* = 0.02). The association between PPI use and ID lost significance when we additionally adjusted for other medication use (OR: 1.43; 95% CI 0.96–2.12, *p* = 0.08). In further models, in which we adjusted separately for each type of medication, it appeared that mainly diuretic use contributed to the attenuation of the association (Appendix A). Associations of all potential confounders with ID are provided in Appendix A. These analyses demonstrated that besides PPI use, also female sex, proteinuria, time since transplantation, diuretics use, and CNI use were independently associated with ID.

### 3.4. Dose-Response Analyses

In this study, 237 RTR received a low PPI dose (≤20 mg omeprazole Eq/d) and 126 RTR received a high PPI dose (>20 mg omeprazole Eq/d). As shown in Table 4 and Figure 1, the point estimate of the odds ratio in RTR taking a high PPI dose as compared to RTR taking no PPIs (OR 2.30; 95% CI 1.46–3.62, *p* < 0.001) was higher than in RTR taking a low PPI dose (OR:1.78; 95% CI 1.21–2.62, *p* = 0.004). After adjustment for potential confounders, PPI use remained associated with ID in patients taking a high PPI dose (OR: 1.73, 95% CI 1.05–2.86, *p* = 0.03), but not in RTR taking a low PPI dose (OR: 1.29, 95% CI 0.84–1.98, *p* = 0.25), as shown in Table 4.

### 3.5. Sensitivity Analyses for Risk of ID 

In sensitivity analyses, H2RA users (n = 20) were excluded from analyses (Appendix A). The association between PPI use and risk of ID remained materially unchanged when H2RA users were excluded (OR: 1.99, 95%CI 1.39–2.86, *p* < 0.001). Moreover, the association between PPI use and ID became slightly stronger when the alternative definition of ID (TSAT < 20% and ferritin < 100 µg/L) was used (OR: 2.90, 95% CI 1.94–4.35, *p* < 0.001), and remained significant independent of adjustment for potential confounders (Appendix A).

### 3.6. Description of Excluded RTR Receiving Oral Iron Supplementation

Baseline differences between RTR with oral iron supplementation and without oral iron supplementation are described in the supplemental results and are demonstrated in Appendix A.

## 4. Discussion

In this study, we demonstrate that PPI use is associated with lower iron status and ID in a large cohort of stable RTR. Remarkably, the association between PPI use and risk of ID remained independent of adjustment for important potential confounders, and appeared to be independent of dietary iron intake, a finding that has not been shown previously. Furthermore, we found that RTR using a high PPI dose have a higher risk of ID. These results indicate that PPI use possibly contributes to the high burden of post-transplantation ID in RTR.

During the past few years, several case reports have demonstrated a relationship between PPI use and the occurrence of IDA [11,26]. Recently, these findings have been strengthened by two large population based cohort studies demonstrating an increased risk of ID among subjects from the general population [8,9]. Lam et al. were the first to observe in a large population that chronic use of both PPIs and H2RAs was associated with an increased risk of ID (adjusted OR: 2.49 for PPI use and 1.58 for H2RA use) [9]. A recent study in a large U.K. population found that the risk of ID was 3.6 times higher in subjects using PPIs for at least one year continuously, i.e., with time gaps between PPI prescriptions of less than 30 days [8]. Consistent with our findings, both studies found a positive dose-response relationship, which suggests a potential causal effect of PPIs. However, compared to these studies, the adjusted odds ratios in our study were lower. This may in part be explained by a higher predisposition of ID in RTR compared to subjects from the general population. ID is highly prevalent after renal transplantation and the etiology is multifactorial. For example, high hepcidin and interleukin-6 levels as a result of inflammatory conditions after transplantation may lead to lower intestinal iron uptake due to the down regulation of the ferroportin transporter responsible for iron transport across the enterocyte [13,27,28]. Furthermore, insufficient iron stores at time of transplantation, per-operative blood loss, and inadequate intake of vegetables rich in iron may add to the risk of ID in RTR [29]. Another potential explanation for the lower odds ratio found in our study might be the relative high incidence of the CYP2C19*17 variant in Caucasian populations, which results in ultra-rapid metabolism of PPIs in the liver [30]. Therefore, the association between PPI use and ID might be more pronounced in populations with lower incidences of this CYP polymorphism, such as Asian populations in which the slow metabolizer phenotype is more common [31]. Interestingly, the present study shows that PPI therapy also appears to be an important risk factor of post-transplantation ID. Since this is a modifiable risk factor, we think this finding is worth discussing given that clinicians may not be aware of the additional risk that PPI use constitutes in RTR.

In contrast to our study, no association was found between PPI use and ID in a cohort study of patients with Zollinger–Ellison syndrome [32]. However, these results cannot simply be extrapolated to other populations and it is likely that the negative effect of PPIs on enteric iron absorption may be less pronounced in patients with gastric acid hypersecretion. In another study among 34 patients with primarily reflux esophagitis, there was also no clear evidence that chronic PPI therapy lead to decreased levels of serum iron and ferritin [33].

Several mechanisms by which PPI use may induce ID are proposed in literature. The main mechanism postulated is decreased intestinal absorption of dietary non-heme iron as a consequence of reduced gastric acid secretion by PPIs [34]. In contrast to the absorption of heme iron, dietary non-heme iron is highly dependent on gastric acid to enhance its absorption [13]. Non-heme iron remains soluble as long as the environment remains acidic and is reducing, of which the latter is necessary to form ferrous iron. It has been shown that in an environment with a pH above 2.5 absorption fails [35]. This theory is supported by a study from Hutchinison et al., who demonstrated that absorption of non-heme iron was lower in patients with hereditary hemochromatosis after the use of PPIs for seven days [34].

However, other factors by which PPIs could affect iron absorption are reported. For example, vitamin C is known to facilitate non-heme iron absorption, since it is a strong reducing agent. Secretion of vitamin C by gastric cells is dependent on intragastric pH and decreased bioavailability of vitamin C has been demonstrated in *Helicobacter pylori* positive and negative subjects after 28 days of omeprazole administration [36]. A low vitamin C intake may add to this, which may be a consequence of patients being over-correct in avoiding all citrus juices, while they actually only need to avoid pomelo containing juices to avoid interaction with CNI use [37]. This suggestion is corroborated by the fact that we recently found that vitamin C depletion is very common in RTR [38]. Moreover, interactions between the gut microbiome and iron bioavailability are reported in literature [39,40,41]. It is known that PPIs tremendously alter the composition of the gut microbiome, which may potentially affect intestinal iron absorption [42]. It is furthermore known that 50% of patients do not take their long-term therapy for chronic diseases as prescribed [43]. This is a further unknown factor that could interact with the results and could weaken the associations that we found.

To our knowledge, the influence of iron intake on the association between PPI use and ID has not been previously investigated. Lam and colleagues argued that possibly only subjects with low-normal iron levels or with a low dietary iron intake may become iron deficient [9]. In the present study, the association between PPI use and ID remained unchanged after adjustment for dietary iron intake, which shows that the association between PPI use and ID is not confounded by iron intake. Besides PPI use, we also found that female sex, proteinuria, time since transplantation, diuretics use, and CNI use were independently associated with ID. To date no evidence has been found linking diuretic use to ID [44]. However, it cannot be excluded that diuretics adversely affect iron status via decreased tubular reabsorption, resulting in increased urinary excretion of iron. The same accounts for proteinuria, which is a sign of either glomerular damage, tubular damage, or both. This may lead to increased protein filtration, decreased protein reabsorption, or both, which may result in increased urinary loss of transferrin-bound iron [45]. We also found that CNI use was independently associated with ID. In a previous study among liver transplant recipients, erythropoietin production and hematocrit levels were significantly reduced in CNI users, however the association with iron deficiency was not investigated [46].

Our study has some limitations. First, our study is cross-sectional in nature, and therefore we cannot assume causality. Next to that, we cannot exclude the possibility that the observed association between PPI use and decreased iron status parameters and ID are caused by residual confounding or indication bias. However, several analyses were performed to decrease this possibility. As described above, adjustment for iron intake and history of gastritis or peptic ulcer disease, conditions that can be the cause of a lower iron status, did not change the association between PPI use and ID. Moreover, in sensitivity analyses we excluded RTR using H2RAs and performed logistic regression analyses using an alternative definition of ID, which did not materially change the association. However, when we adjusted for medication use in model 6, the association between PPI use and ID lost significance, which could possibly mean two things. First, other medications may also negatively affect iron status, which attenuates the effect of PPIs on ID. Second, RTR using other medications may be more prone to ID compared to non-users. Furthermore, soluble transferrin receptor (sTfR) measurements were not available, and it should be realized that other than recording history of overt GI-disorders, patients were not thoroughly screened for presence of GI-disorders at the moment of sampling. Lastly, this is a single center study consisting of predominantly Caucasian RTR, which may limit generalizability of results to other populations.

Our study also has several strengths. It is the first study in its kind investigating the association of PPI use with iron status parameters and ID in a large cohort of stable RTR. The main strength is the well-characterized cohort of RTR, in which multiple iron status parameters and dietary iron intake were measured. Extensive data collection made it possible to correct for many possible confounding factors, including lifestyle parameters, inflammation, and medication use. Lastly, to define ID we used a definition previously used in chronic kidney disease and in RTR, including both functional and absolute iron deficiency situations [3,17,18].

## 5. Conclusions

In conclusion, we demonstrated that PPI use is associated with lower iron status and ID, indicating impaired intestinal absorption of iron potentially related to reduced gastric acid secretion. Moreover, we demonstrated that the association was stronger among RTR taking a high PPI dose. Taken together these results infer that PPI use is an important modifiable factor that potentially contributes to the high burden of post-transplantation ID after renal transplantation. Based on these results it should be advised to actively manage iron status in RTR using chronic PPI therapy. Reevaluation of treatment indication or switching to a less potent acid suppressing drug, such as antacids or H2RAs, might also be considered in RTR with ID. Potential clinical consequences associated with ID underscore this, including premature mortality [3] and severely disabling restless legs syndrome, which has a reported prevalence of 51.5% among RTR [47]. Since the majority of studies investigating the association between PPI use and ID are observational, randomized controlled clinical trials are needed to determine a causal effect of PPI use on iron status.

## Figures and Tables

**Figure 1 jcm-08-01382-f001:**
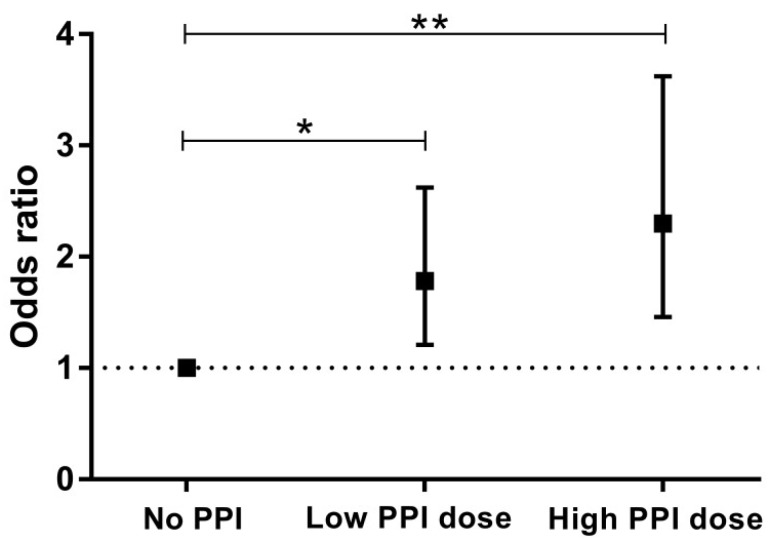
Crude association between PPI use and risk of iron deficiency stratified by subgroups of PPI use. No PPI, low PPI dose (≤20 mg omeprazole Eq/d), high PPI dose (>20 mg omeprazole Eq/d). Presented are odds ratio’s with 95% confidence intervals. ** and * represent significant *p* values compared to No PPI subgroup.

**Table 1 jcm-08-01382-t001:** Baseline characteristics of 646 renal transplant recipients.

Characteristics		Total Population	Non-PPI User	PPI User	*p*
Number of subjects, n (%)		646 (100)	283 (43.8)	363 (56.2)	n/a
Demographics					
	Age, years	53 ± 13	51 ± 13	54 ± 12	0.001
	Men, n (%)	382 (59.1)	170 (60.1)	212 (58.4)	0.7
	BMI, kg/m^2^	26.7 ± 4.8	26.0 ± 4.6	27.3 ± 4.8	<0.001
	Diabetes Mellitus, n (%)	157 (24.3)	54 (19.1)	103 (28.4)	0.006
	History of gastrointestinal disorders, n (%)	42 (6.5)	10 (3.5)	32 (8.8)	0.007
	Time since transplantation, years	5.3 (1.8–12.0)	9.5 (4.1–15.0)	4.0 (1.1–8.0)	<0.001
Lifestyle parameters					
	Current smoker, n (%)	79 (13.1)	33 (12.4)	46 (13.6)	0.7
	Alcohol consumer, n (%)	409 (70.6)	186 (72.7)	223 (69.0)	0.3
	Iron intake, mg/d	11.3 ± 2.9	11.2 ± 2.7	11.4 ± 3.0	0.5
Renal function parameters					
	eGFR, mL/min/1.73 m^2^	53.5 ± 19.9	56.2 ± 19.7	51.4 ± 19.8	0.002
	Serum creatinine, µmol/L	122 (99–156)	117 (98–150)	126 (101–164)	0.03
	Proteinuria (≥0.5 g/24 h), n (%)	135 (21.0)	60 (21.2)	75 (20.8)	0.9
Laboratory parameters					
	Iron deficiency, n (%)	193 (29.9)	63 (22.3)	130 (35.8)	<0.001
	Hb, g/dL	13.3 ± 1.7	13.6 ± 1.6	13.1 ± 1.8	<0.001
	Iron, µmol/L	15.2 ± 5.9	16.4 ± 6.1	14.2 ± 5.6	<0.001
	Ferritin, µg/L	115.5 (53.0–216.3)	136.0 (77.0–222.0)	93.0 (42.0–196.0)	<0.001
	Transferrin saturation, %	25.1 ± 10.5	27.3 ± 10.1	23.3 ± 10.5	<0.001
	Glucose, mmol/L	5.3 (4.8–6.0)	5.2 (4.7–5.8)	5.3 (4.9–6.2)	0.01
	HbA1c, mmol/mol	40 (37–44)	39 (36 – 42)	41 (38 – 45)	<0.001
	HsCRP, mg/L	1.6 (0.8–4.2)	1.6 (0.8–3.8)	1.6 (0.7–4.6)	0.8
Medication use					
	Calcineurin inhibitors, n (%)	369 (57.1)	137 (48.4)	232 (63.9)	<0.001
	Mycophenolate mofetil, n (%)	431 (66.7)	171 (60.4)	260 (71.6)	0.003
	Prednisolone, n (%)	641 (99.2)	282 (99.6)	359 (98.9)	0.4
	Diuretics, n (%)	253 (39.2)	87 (30.7)	166 (45.7)	<0.001
	RAAS–inhibitors, n (%)	314 (48.6)	144 (50.9)	170 (46.8)	0.3
	Antiplatelet drugs, n (%)	131 (20.3)	46 (16.3)	85 (23.4)	0.03
	H2-receptor antagonists, n (%)	20 (3.1)	19 (6.7)	1 (0.3)	<0.001

Data are presented as mean ± SD, median with interquartile ranges (IQR) or number with percentages (%). Abbreviations: BMI, body mass index; eGFR, estimated glomerular filtration rate; Hb, hemoglobin; HbA1c, hemoglobin A1c; HsCRP, high-sensitivity C-reactive protein; PPI, proton-pump inhibitor; RAAS-inhibitors, renin-angiotensin-aldosterone system inhibitors.

**Table 2 jcm-08-01382-t002:** Association of PPI use with iron status parameters in 646 stable renal transplant recipients.

	Serum Iron, µmol/L			Ln Serum Ferritin, µg/L			Transferrin Saturation, %			Hemoglobin, g/dL		
n = 646	β	95% CI	*p*	β	95% CI	*p*	β	95% CI	*p*	β	95% CI	*p*
Crude	−2.18	−3.09; −1.27	<0.001	−0.34	−0.49; −0.18	<0.001	−3.92	−5.52; −2.32	<0.001	−0.52	−0.78; −0.25	<0.001
Model 1	−2.03	−2.94; −1.12	<0.001	−0.35	−0.50; −0.20	<0.001	−3.80	−5.40; −2.20	<0.001	−0.52	−0.78; −0.26	<0.001
Model 2	−1.61	−2.57; −0.65	0.001	−0.31	−0.48; −0.15	<0.001	−2.85	−4.55; −1.15	0.001	−0.35	−0.61; −0.10	0.007
Model 3	−1.67	−2.67; −0.66	0.001	−0.31	−0.48; −0.14	<0.001	−3.00	−4.80; −1.20	0.001	−0.41	−0.67; −0.14	0.003
Model 4	−1.54	−2.48; −0.60	0.001	−0.32	−0.48; −0.16	<0.001	−2.75	−4.43; −1.07	0.001	−0.35	−0.61; −0.09	0.007
Model 5	−1.62	−2.58; −0.66	0.001	−0.31	−0.47; −0.15	<0.001	−2.90	−4.60; −1.20	0.001	−0.35	−0.61; −0.01	0.007
Model 6	−1.37	−2.33; −0.41	0.005	−0.27	−0.43; −0.11	0.001	−2.33	−4.03; −0.63	0.007	−0.33	−0.58; −0.07	0.01

Model 1: PPI use adjusted for age and sex. Model 2: model 1 + adjustment for eGFR, proteinuria, time since transplantation, history of GI-disorders. Model 3: model 2 + adjustment for lifestyle parameters (BMI, smoking behavior, alcohol use, dietary iron intake). Model 4: model 2 + adjustment for inflammation (hs-CRP). Model 5: model 2 + adjustment for MMF use. Model 6: model 5 + adjustment for other medication use (diuretic use, RAAS-inhibition, antiplatelet therapy, CNI use, and prednisolone use). Abbreviations: CNI, calcineurin inhibitor; Ln, natural log transformed; MMF, mycophenolate mofetil; RAAS-inhibitors, renin-angiotensin-aldosterone system inhibitors.

**Table 3 jcm-08-01382-t003:** Logistic regression analyses investigating the association of PPI use with iron deficiency in 646 renal transplant recipients.

	Iron Deficiency		
n = 646	Odds Ratio	95% CI	*p*
**Crude**	1.95	1.37–2.77	<0.001
**Model 1**	1.94	1.36–2.78	<0.001
**Model 2**	1.57	1.07–2.31	0.02
**Model 3**	1.57	1.04–2.38	0.03
**Model 4**	1.56	1.06–2.30	0.03
**Model 5**	1.57	1.07–2.31	0.02
**Model 6**	1.43	0.96–2.12	0.08

Model 1: PPI use adjusted for age and sex. Model 2: model 1 + adjustment for eGFR, proteinuria, time since transplantation, history of GI-disorders. Model 3: model 2 + adjustment for lifestyle parameters (BMI, smoking behavior, alcohol use, dietary iron intake). Model 4: model 2 + adjustment for inflammation (hs-CRP). Model 5: model 2 + adjustment for MMF use. Model 6: model 5 + adjustment for other medication use (diuretic use, RAAS-inhibition, antiplatelet therapy, CNI use, and prednisolone use). Abbreviations: CNI, calcineurin inhibitor; MMF, mycophenolate mofetil; RAAS-inhibitors, renin-angiotensin-aldosterone system inhibitors.

**Table 4 jcm-08-01382-t004:** Subgroup analyses of the association of PPI use with iron deficiency in 646 stable renal transplant recipients.

	Categories of PPI Use						
	**No PPI**		**Low PPI Dose**		**High PPI Dose**		*p* trend
**Number of subjects**	283		237		126		
	Odds ratio (95% CI)	*p* value	Odds ratio (95% CI)	*p* value	Odds ratio (95% CI)	*p* value	
**Iron deficiency**							
Crude	1.00 (reference)	n/a	1.78 (1.21–2.62)	0.004	2.30 (1.46–3.62)	<0.001	<0.001
Model 1	1.00 (reference)	n/a	1.76 (1.19–2.62)	0.005	2.33 (1.47–3.69)	<0.001	<0.001
Model 2	1.00 (reference)	n/a	1.38 (0.90–2.10)	0.14	2.00 (1.23–3.25)	0.005	0.005
Model 3	1.00 (reference)	n/a	1.43 (0.91–2.24)	0.12	1.88 (1.11–3.16)	0.02	0.02
Model 4	1.00 (reference)	n/a	1.39 (0.91–2.13)	0.12	1.93 (1.18–3.15)	0.009	0.008
Model 5	1.00 (reference)	n/a	1.38 (0.91–2.10)	0.14	2.00 (1.23–3.26)	0.005	0.005
Model 6	1.00 (reference)	n/a	1.29 (0.84–1.98)	0.25	1.73 (1.05–2.86)	0.03	0.03

Model 1: PPI use adjusted for age and sex. Model 2: model 1 + adjustment for eGFR, proteinuria, time since transplantation, history of GI-disorders. Model 3: model 2 + adjustment for lifestyle parameters (BMI, smoking behavior, alcohol use, dietary iron intake). Model 4: model 2 + adjustment for inflammation (hs-CRP). Model 5: model 2 + adjustment for MMF use. Model 6: model 5 + adjustment for other medication use (diuretic use, RAAS-inhibition, antiplatelet therapy, CNI use, and prednisolone use). Abbreviations: CNI, calcineurin inhibitor; MMF, mycophenolate mofetil; RAAS-inhibitors, renin-angiotensin-aldosterone system inhibitors.

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
