# Peer review of "Chronic Use of Proton-Pump Inhibitors and Iron Status in Renal Transplant Recipients"

_jcm, 2019, doi:10.3390/jcm8091382_

Round 1
Reviewer 1 Report
Douwes et al. adressed all major points thoroughly and now present an interesting paper fit for publication. After detailed examination of the manuscript and comments I have nothing to add.
Reviewer 2 Report
All my comments of te first review were integrated. The article is ready to be published.
This manuscript is a resubmission of an earlier submission. The following is a list of the peer review reports and author responses from that submission.
Round 1
Reviewer 1 Report
This is the first study in its kind investigating the association of proton-pump inhibitor (PPI) use with iron status parameters and iron deficiency (ID) in a large single-center cohort of stable outpatient renal transplant recipients (RTR) with well characterized cohort. This study considered dietary iron intake and H2-receptor antagonists influence on ID.
However, several points remain unclear:
1) Iron deficiency was defined as transferrin saturation (TSAT) 107 <20% and ferritin <300 μg/L as described in literature (106-107 lines). Based on one reference.
Regarding NIH following blood test to diagnose iron-deficiency anemia might be perfored: complete blood count, iron level measurement, ferritin measure, reticulocyte count, peripheral smear. Ferritin level less than 10 μg/L show ID and iron level less than 10 show ID for both men and women.
Methodology requires consideration, as well data presented in research shows that ferritin level differed from 42-222 μg/L, iron mean level was between 14-16 μmol/L.
2) The association between PPI use and ID lost significance when we additionally adjusted for medication use (OR: 1.43; 95%CI 0.96-2.12, P=0.08) (172 line).
This part should be reconsidered taking into account medication separately. Common adverse effect of immunosuppressant mycophenolate mofetil (MMF) is anemia which reflect the immunosuppressive and myelosuppressive nature of the drug. Besides PPI are basically prescribed due to MMF induced complications (gastrointestinal disturbances). If MMF is prescribed together with PPI, MMF should correlate to ID as well. Decision either ID is related to PPI or either induced by MMF should be reviewed.
Moreover, clinical relevance in discussion section might be useful. What can protect from PPI induced anemia? Should these medication be switched to H2-receptor antagonists? Any advice for clinicians might attract more readers.
Reviewer 2 Report
This is a very well written paper, with important news for the clinical practice.
I reviewed the paper from a nursing perspective. Here I have some suggestion to improve the discussion of the paper.
In our hospital we educate transplanted patients to not take pomelo juice as it could interact with Immunosuppressive Therapy. As the literature showed, all the other citrus juices do not exhibit significant interaction (Sridharan & Sivaramakrishnan, 2016). Our patient however, to be over correct avoided all juices. This could be a factor tob e taken into consideration with the association of Vitamin C and iron (Line 245).
Secondly, some of our patient take „healing earth digest calm“ (Heilerde) against irritable stomach. It is known that 50% of chronically-ill patients do not take pharmaceutical drugs as prescribed (Pagès-Puigdemont u. a., 2016). Here we have a further unknown factor that could interact with the results.
Finally Willis-Ekbom Disease (https://www.ncbi.nlm.nih.gov/pmc/articles/PMC4627182/) in the Article “Sleep apnea syndrome and restless legs syndrome in kidney transplant recipients” showed that 51,5% of transplanted kidney recipient had symptoms of RLS. As RLS is associated with iron deficiency the clinical consequence in practice is big.
RLS has been reported to be particularly frequent in association with some clinical conditions such as peripheral neuropathy, type 2 diabetes, uremia, multiple sclerosis, iron deficiency, hypothyroidism or hyperthyroidism, acute intermittent porphyria, and pregnancy.[26,27,28,29]
26. Walters AS. The International Restless Legs Syndrome study group. Toward a better definition of the restless legs syndrome. Mov Disord. 1995;10:634–42.
27. Allen RP, Picchietti D, Hening WA, Trenkwalder C, Walters AS, Montplaisi J. Restless legs syndrome: Diagnostic criteria, special considerations, and epidemiology. A report from the restless legs syndrome diagnosis and epidemiology workshop at the National Institutes of Health. Sleep Med. 2003;4:101–19.
28. Ulfberg J, Nystrom B, Carter N, Edling C. Prevalence of restless legs syndrome among men aged 18 to 64 years: An association with somatic disease and neuropsychiatric symptoms. Mov Disord. 2001;16:1159–63.
29. Tachibana N, Tanigawa T. Prevalance and clinical characteristics of restless legs syndrome among Japanese industrial workers. Neurology. 2003;60:38.
Reviewer 3 Report
Douwes et al. present an interesting study on a underestimated side effect of frequently used PPI. Iron deficiency anemia is a common finding in transplant recipients and is associated with bad outcomes. The authors summarized the background and study rationale very well in the introduction.
The study seems well designed and all ethical requirements appear to be fulfilled. The following major issues need to be addressed:
Iron deficiency was defined by a ferritin concentration <300 µg/L and TSAT of <20%. Outside the field of nephrology this cut-off values are too high. Considering the fact that most patients have normal kidney function, the authors need to explain why not more conservative cut-off values were used. The cut-off concentration of ferritin that is diagnostic for iron def. varies between 12 and 15 μg/l unless there is coexistent inflammatory disease.
If available, sTfR/log10 serum ferritin ratio should be included to differentiate between true iron def. and functional iron def.
PPI are often prescribed due to upper and lower gastrointestinal diseases (eg. oesophagitis, erosions and peptic ulcer disease), conditions which can cause iron def. by itself. Have patients been thoroughly investigated for this major confounder? How strong is the association of iron def. and history of GI disorders?
In the logistic regression analysis, the associations of the all the confounders are not shown and could be interesting.
How many patients received oral or intravenous iron treatment since transplantation? How did those patients compare to patients without iron suppl. And PPI intake.